# Imbalance and Falls in Patients with Parkinson’s Disease: Causes and Recent Developments in Training and Sensor-Based Assessment

**DOI:** 10.3390/brainsci14070625

**Published:** 2024-06-22

**Authors:** Veit Mylius, Elisabeth Zenev, Caroline S. Brook, Florian Brugger, Walter Maetzler, Roman Gonzenbach, Anisoara Paraschiv-Ionescu

**Affiliations:** 1Department of Neurology, Center for Neurorehabilitation, 7317 Valens, Switzerland; eli.zenev@gmx.de (E.Z.); carobrook@hotmail.com (C.S.B.); roman.gonzenbach@kliniken-valens.ch (R.G.); 2Department of Neurology, Philipps University, 35043 Marburg, Germany; 3Department of Neurology, University of Bern, Inselspital Bern, 3010 Bern, Switzerland; 4Department of Neurology, Kantonsspital St. Gallen, 9007 St. Gallen, Switzerland; florian.brugger@kssg.ch; 5Department of Neurology, University Hospital Schleswig-Holstein, Kiel University, 24105 Kiel, Germany; w.maetzler@neurologie.uni-kiel.de; 6Signal Processing Laboratory 5, Ecole Polytechnique Federale de Lausanne (EPFL), 1015 Lausanne, Switzerland; anisoara.ionescu@epfl.ch

**Keywords:** Parkinson’s disease, clinical research, technology, imbalance, falls

## Abstract

Imbalance and falls in patients with Parkinson’s disease (PD) do not only reduce their quality of life but also their life expectancy. Aging-related symptoms as well as disease-specific motor and non-motor symptoms contribute to these conditions and should be treated when appropriate. In addition to an active lifestyle, advanced exercise training is useful and effective, especially for less medically responsive symptoms such as freezing of gait and postural instability at advanced stages. As treadmill training in non-immersive virtual reality, including dual tasks, significantly reduced the number of falls in PD patients, the mechanism(s) explaining this effect should be further investigated. Such research could help to select the most suitable patients and develop the most effective training protocols based on this novel technology. Real-life digital surrogate markers of mobility, such as those describing aspects of endurance, performance, and the complexity of specific movements, can further improve the quality of mobility assessment using wearables.

## 1. Introduction

Gait disturbances and falls are common with age, leading not only to reduced participation and quality of life, but also to reduced life expectancy, mainly due to immobility caused by fractures [1,2]. An initiative of the Centers for Disease Control and Prevention (CDC) in the US proposed a yearly screening using the Stopping Elderly Accidents, Deaths and Injuries (STEADI) brochure for patients with falls. Their algorithm considers risk factors such as polypharmacy and osteoporosis as well as the examination of vision, mental health, hypotension, and gait [3,4]. Further factors contributing to falls can be summarized under frailty (including weakness, low gait speed, low physical activity, unintended weight loss, and self-reported exhaustion, but also morbidity), leading to a sedentary lifestyle further increasing the risk of falls [5,6]. In addition, cognitive decline reduces gait velocity and increases imbalance probably due to more pronounced trade-off effects [7,8]. In patients with Parkinson’s disease (PD), various motor and/or non-motor symptoms additionally contribute to a walking disorder with imbalance [9,10]. Here, recurrent falls occur in 42% after a disease duration of 8 years and in 72% after 16 years [11]. Therefore, a comprehensive diagnostic by the general practitioner as well as by a specialist is required, addressing aging and PD associated causes as detailed below. The aim of therapy should be to improve daily functioning by treating risk factors for falls using pharmacological and non-pharmacological approaches. The latter are justified not only as a prophylactic treatment for age-related frailty, but also for symptoms that are difficult to treat with medication, such as fear of falling, freezing of gait (FoG), and postural instability.

Virtual reality (VR) holds promise for enhancing the understanding and treatment of complex impairments in PD by immersing individuals in enriched and highly personalized environments that simulate real-world scenarios while minimizing training risks. Yet, the full potential of VR in PD rehabilitation remains unrealized [12]. Currently, there is limited evidence supporting the superiority of VR-based rehabilitation over non-VR approaches in improving gait and balance outcomes, although both are more effective than no intervention when provided under full supervision. VR presents opportunities to safely identify specific triggers for FoG and balance deficits, thereby guiding personalized training objectives. To harness the potential of VR rehabilitation and optimize treatment outcomes, researchers are urged to develop immersive VR applications incorporating integrated assessment and training modules tailored to the needs of individuals with PD and healthcare providers.

Wearable motion sensors, due to their compact size, light weight, and low power consumption, have proven clinically valuable in healthcare and daily-life monitoring. Accelerometers and gyroscopes are commonly integrated into an inertial measurement unit (IMU), enabling the capture of three-dimensional linear acceleration and angular velocities of body segments. When paired with advanced and validated algorithms, wearable systems allow estimating a range of gait and physical activity parameters in real-life conditions, by providing insights into aspects such as gait speed, stride length, cadence, and intensity and patterns of physical activity [13,14]. This potential of a comprehensive characterization of motor behavior presents the opportunity to enhance the assessment of PD motor symptoms by monitoring free-living movements over extended periods outside the lab/clinical setting. Research suggests that PD patients often walk better under observation compared to unsupervised daily activities [15]. Free-living activities involve various tasks with different challenges and distractions, potentially reducing attention and increasing the risk/fear of falling. Moreover, certain PD-related episodes, such as the on/off phenomenon and FoG, may be challenging to detect during lab-based observation due to their complexity or rarity. Therefore, there is clinical consensus that a comprehensive evaluation of PD patients requires data collection over extended observation periods while patients engage in their normal daily activities, along with a deeper understanding of the significance of sensor-derived mobility parameters compared to clinical data [16].

This review summarizes potential causes for imbalance in patients with PD and discusses recent advances in mobility assessment using wearable devices, as well as gait training, including technologies such as virtual reality-based gait training.

## 2. Causes for Imbalance and Falls in Parkinson’s Disease

In mild PD, especially fear of falling but also a history of near falls and retropulsion (probably due to postural instability in most patients) were found to be independent factors contributing to subsequent falls [17]. In PD patients with falls, >2 falls in the past year, the presence of motor fluctuations, a Unified PD Rating Scale (UPDRS) activities of daily living (ADL) score > 12, Levodopa equivalent daily doses (LEDD) > 700 mg, and a Berg balance scale score of <49 points predict subsequent falls with moderate to high accuracy [18]. These data show that there is a large inter-individual variance in the risk of falling, strongly arguing in favor of a comprehensive individual risk assessment in each patient (see Table 1). Another comprehensive approach summarizing the risk factors for falls in PD can be found elsewhere [19]. Relevant factors in PD patients beyond aging (for details see Introduction) are detailed as follows. 

### 2.1. Motor Symptoms Associated with PD and Falls

Motor symptoms leading to falls in patients with PD are bradykinesia, rigidity, motor fluctuations (Off-phases and/or choreatic dyskinesia in the On-phase), freezing of gait (FoG), and postural instability. Bradykinesia and rigidity result in smaller movement amplitudes, which is reflected, for example, in decreased gait velocity, shorter step length, and asymmetry of gait. Postural deformities further contribute to imbalance by changing the center of gravity. As the disease progresses, postural instability and FoG in particular may contribute. In advanced stages of the disease, when the therapeutic window of levodopa is narrowing, motor fluctuations, especially off periods (e.g., wearing off and early morning off), are critical times of the day with an increased risk of falls [36]. However, choreatic dyskinetic phases can also negatively affect balance and lead to falls. These motor fluctuations require an optimized dopaminergic treatment regime adapted to patient’s diaries and the assessment of motor function. With disease progression, pharmacological treatment options of motor fluctuations, FoG, and postural instability are often limited, requiring additional physiotherapeutic approaches.

### 2.2. Non-Motor Symptoms Associated with PD and Falls

Non-motor symptoms associated with imbalance and falls in PD are orthostatic hypotension, cognitive deficits, anxiety, and depression, as well as sensory disturbances [18,37,38]. They are often underdiagnosed, and dopaminergic but also non-dopaminergic as well as non-pharmacological treatment should be consequently introduced when appropriate [39,40,41]. Orthostatic hypotension is considered a prevalent and extremely relevant factor, with a need for specialized diagnostics and treatment [9]. Standardized blood pressure monitoring as well pharmacological and non-pharmacological approaches should be introduced [42,43,44]. Reduced cognitive-executive function, especially reduced motor and cognitive dual-task abilities, were assumed to be main drivers for falls [45,46]. Both motor and cognitive dual-task abilities were reduced in dual-task walking in PD patients with mild cognitive impairment, resulting in a wrong prioritization of the tasks (“posture second”) [46]. Thus, the training of dual-task capacity, allowing patients to control conflicting situations, should be introduced for these patients (see next section). Anxiety and depression in PD are associated with fear of falling, which further reduces physical activity, leading to imbalance and falls [9,17]. Sensory impairments such as visual impairment, altered sensation, and chronic pain affect balance through reduced ability to negotiate obstacles and impaired equilibrium. Chronic pain also reduces physical activity, which can increase depression and anxiety, further affecting balance [47].

### 2.3. Factors Indirectly Associated with PD and Falls

In addition, postural deformities with orthopedic consequences and polyneuropathy (often subclinical), which are more common in this population, can both negatively affect balance [43,48]. For example, there is evidence that PD patients with back pain presented with more pronounced thoraco-lumbar kyphosis correlating with disease progression [35]. On the other hand, continuous or prolonged high-dose dopaminergic therapy may decrease cobalamin levels, leading to polyneuropathy (and/or affection of the central afferents) [34], already observed in patients without clinical impairment [33]. Therefore, the individual management of PD-related gait and balance deficits associated with falls must take into account a range of age- and PD-related factors (Table 1). Once deficits become apparent, patients should receive advanced gait and balance training as soon as possible, to prevent further complications, as discussed below.

## 3. Advanced Gait and Balance Training in Parkinson’s Disease

In PD, gait training is strongly advised not only to reduce frailty aspects in this vulnerable cohort, but also to improve motor and non-motor symptoms that cannot be adequately managed using medication. Convincing effects on various outcomes were seen for treadmill training (TT) and home-trainer-based endurance training, dual-task training, cognitive training for freezing, and combined approaches such as virtual-reality-enhanced treadmill training (VR-TT) (see Table 2). Hence, the relevance of clinical and laboratory-based outcomes should be further explored using wearable devices in daily living [16].

In early PD, both high-intensity TT four times/week and home-trainer based aerobic exercise three times/week for 6 months showed relatively constant UPDRS motor scores as compared to the worsening in the control groups [49,50]. At-home physiotherapeutic controlled gait training together or apart with cognitive tasks for 6 weeks both led to faster dual-task gait velocities, which was retained at the 12-week follow-up [51]. Also, cognitive training twice weekly for 7 weeks reduced the percentages of FoG during the TUG test [52]. According to individual preferences and to increased compliance, Tai Chi over 4 years (effects on UPDRS and LEDD in an uncontrolled study), Mindfulness Yoga for 8 weeks (positive effects especially on mood and UPDRS III), and rhythmic auditory stimulation (RAS) combined with balance training over 5 weeks (effects on mobility (Mini-BESTest)) even in patients with Hoehn and Yahr stage IV) can be recommended, since convincing effects on various outcomes were shown [54,55,56]. With respect to falls, combined dual-task training in enhanced virtual-reality integrated into TT revealed marked effects in a mixed population of elderly patients and patients with PD and falls [53]. Their trial compared falls 6 months before and after VR-TT or TT alone three times a week for 6 weeks and showed a halving of falls, with even stronger effects in a subgroup of Parkinson’s patients. Aging and PD-associated falls were only reduced significantly by VR-TT, but not by TT alone. A secondary analysis from a previous study showed that patients with FoG profit to a similar extend but without reducing FoG [57] (for study design and outcomes, see Table 2).

Apart from training physical capacity, training to perform dual tasks is thought to contribute to the effects of VR-TT: functional imaging of these PD patients yielded fMRI activity changes in cortical areas involved in frontal executive function as well as cerebellar activity [58]. Frontal activations were interpreted as compensatory, whereas cerebellar activity changes may reflect effects on automaticity. Further observations showed an implication of a different striatal connection compared to those usually involved in cognitive and motor dual tasks [59]. This shift in brain activity may explain the non-favorable prioritization of PD patients with cognitive impairment (posture second), as their capacity to perform cognitive and motor tasks in parallel is reduced [46].

Taken together, these observations suggest that the dual-task VR-TT approach may improve several causes of falls in the elderly population as well as in patients with PD. It may not only increase physical capacity but also automaticity and especially dual-task capabilities. However, whether this results in the restoration of previous striatal connectivity remains to be investigated. In addition, its positive effect on falls and other outcomes promotes a non-sedentary lifestyle. With the exception of FoG, motor and non-motor symptoms leading to falls have not been controlled for in selected groups of PD patients. It would be of interest to investigate shorter intensive or ambulatory maintenance protocols. Also, the optimal training program (with respect to the dual tasks and their level of difficulty) and intensity (e.g., high-intensity) should be explored to account for the respective cause of falls. Various studies addressing these issues are ongoing (for study protocols, see [60,61]).

In summary, the increase in automaticity and dual-task capabilities by means of dual-task VR-TT may improve and/or compensate for various risk factors for falls (i.e., FoG, postural instability, and frailty). The type and amount of contribution of specific risk factors should be further examined in different settings (i.e., maintenance paradigm or short-term training). Wearable devices, addressing the complexity of gait in daily living, may help to explore training effects in PD patients with falls (e.g., effects on lifestyle).

## 4. Wearable Devices for Gait and Balance Assessment in Parkinson’s Disease

Classical methods of gait analyses include clinical observations, structured clinical assessments, locomotion-specific tests, and more sophisticated, automatized analysis in a gait lab. The use of wearable devices, however, allows investigating gait in a more realistic environment. Recent findings indicate notable differences in gait measurements between clinical or laboratory settings and those observed during everyday activities [62]. Additionally, assessments based on community ambulation appear to offer valuable insights into predicting critical outcomes related to fall risk [63] and quality of life [64]. These insights underscore the importance of integrating assessments of mobility during daily activities, complementing traditional clinic and laboratory evaluations of gait and motor function.

Wearable devices and associated data analytics have the potential to provide parameters reflecting the patient’s real **performance** in daily living and should be an outcome marker in clinical studies, since self- and laboratory-based assessments rather display self-perception and the patient’s potential under direct investigation (**i.e., perception and capacity**) [62,65]. Thus, the gap between laboratory-based capacity (i.e., the potential of a patient) and performance may guide individual therapy to set improvements and to increase compliance [66]. The assessment of performance using wearables also compared to capacity and self-perception is currently under investigation. Despite the clinical significance, long-term monitoring in real-life conditions poses technical and practical challenges. For example, the positioning and number of sensors are critical for assessing various symptoms across different subjects. Motor fluctuations associated with PD impact daily mobility in diverse ways depending on disease stage and clinical phenotype. Therefore, employing multiple devices may be more appropriate for comprehensive evaluations, but this could compromise the comfort and acceptance of PD patients in wearing the system. Moreover, assessing movement using multiple devices, such as gait analysis, requires the precise time synchronization of data streams, a capability offered by only a limited number of commercial IMU-based devices.

Regarding impaired locomotion (e.g., slow walking speed, asymmetrical gait, and movement smoothness), the center of mass has been widely utilized in research to assess movement performance and stability levels. For monitoring activities like walking and turning, consensus in the literature suggests using a single sensor positioned near the waist or lower back [67,68,69].

The gait of PD patients in everyday settings has been investigated using a single wearable device positioned on the lower back for prolonged periods (e.g., 7 days). This examination aimed to assess various aspects of ambulation, such as volume, pattern, and variability, along with temporal and spatial gait characteristics, which were found to differ from controlled laboratory conditions [70]. Findings from multiple studies utilizing the same sensor setup revealed that both the quality and quantity of parameters describing gait and turning were associated not only with fall status (fallers vs. non-fallers) but also with PD-specific characteristics (e.g., PD fallers exhibiting higher variability compared to older adult fallers) [71,72].

Beyond classical gait parameters, the quality of movement control matters. How hesitant, stable, abrupt, or smooth the gait is can provide valuable insights into PD progression, therefore assessing mobility smoothness using robust smoothness measures, such as the spectral arc length (SPARC), may reveal unique characteristics among populations prone to falling. The results of a recent study showed that SPARC measures were significantly lower (indicating less smoothness) in PD patients compared to controls, particularly when derived from acceleration signals [73]. These SPARC measures correlated with the severity of motor symptoms assessed using the Unified Parkinson’s Disease Rating Scale (UPDRS). Additionally, PD patients exhibited smoother gait when medicated compared to when off medication. The findings suggest that SPARC is a valuable tool for quantifying gait smoothness in PD, providing insights into disease progression and treatment effects. While data collection for that study occurred in laboratory settings, the SPARC smoothness index is recognized as a robust measure that accounts for walking bout duration, movement speed, and signal noise artifacts—frequently encountered in accelerometry-based measurements. Thus, it holds promise for analyzing data collected in real-world daily-living conditions.

As gait encompasses multiple characteristics, accurately quantifying various parameters is crucial for identifying pathology and specific disease features. However, due to the high covariance among these characteristics, efforts have been made to develop conceptual gait models to reduce redundancy and enhance interpretation. Data reduction statistical tools, such as principal component analysis and factor analysis, have been explored to identify domains grouping the various gait characteristics. For example, a model has been proposed using wearable data from older adults and those with PD during free living, resulting in 14 gait characteristics across four domains: (i) pace (e.g., step velocity and step length), (ii) rhythm (e.g., step, stance, and swing time), (iii) variability (e.g., variance of step, stance, and swing time), and (iv) asymmetry (e.g., asymmetry of step, swing, and stance time) [74].

In addition to gait parameters, wearable-derived outcomes are dimensions of physical activity (type, intensity, frequency, and pattern). These dimensions can be expressed as **endurance, performance, and complexity [75]. Endurance** includes the total locomotion time, the longest locomotion period, and the usual walking cadence. **Performance** reflects the cadence of the longest locomotion period and locomotion periods with at least 30 steps and 100 steps/min. **Complexity** reflects the ability to react in the natural environment and is quantified according to the type (i.e., sedentary and locomotion), intensity (acceleration and walking cadence), and duration of activity (i.e., distribution of locomotion periods) expressed based on the Lempel–Ziv complexity (LZC) theory [76] (see Figure 1, modified from [75]).

In addition to aging, these parameters are under mutual influence of external and internal factors (e.g., dual-tasking and fear of falling), which need to be further explored to define outcome measures in PD [61]. In the healthy elderly, a single device attached to the trunk showed a decreased complexity of physical activity in participants with concerns about falling, reflecting a part of the vicious cycle of inactivity and falling [75].

## 5. Discussion

With so many contributing factors, managing PD-related gait disturbance with falls remains a challenge. It requires a holistic approach involving patients, their caregivers, the general practitioner (to account of age-related factors), and therapy and treatment specialists, with the aim of participation and self-management (for a review, see [77]). The proposed short-list of aging and PD-related causes for falls may help the clinician to address the most relevant factors for each patient (Table 1). In addition to their treatment, an active lifestyle is encouraged, combined with physiotherapist-led training based on individual challenges [78]. In general, various training approaches have been proposed to compensate for reduced automaticity, such as reward-based learning (to increase dopamine release) and learning new skills [79,80]. Individual training preferences help to increase motivation (also through social interactions), as positive effects have been observed for many different interventions (e.g., Mindfulness Yoga or Tai Chi).

For treatment-resistant PD symptoms such as FoG and postural instability, specific training approaches are warranted. Dual-task training is thought to compensate diminished cognitive-executive function and reaction velocity towards unexpected obstacles. However, the literature on effective dual-task therapies to reduce falls remains scarce. To date, only one trial of 6 weeks of VR-enhanced TT showed convincing effects over 6 months on the frequency of falls in older people and patients with Parkinson’s disease, including those with FoG. Other dual-task interventions also showed short-term effects on relevant outcomes, but without directly addressing falls. For example, rhythmic auditory stimulation (RAS) combined with balance training showed effects on the Mini-BESTest gait test.

The recognition of walking speed as the sixth vital sign has driven significant collaborative research aimed at extensively validating algorithms, both technically and clinically, for estimating walking speed and related parameters (such as cadence and stride length) in challenging conditions marked by real-life contexts and impaired gait patterns (https://mobilise-d.eu/, accessed on 10 April 2024). This opens new perspectives for an objective and quantitative assessment of real-world walking and physical activity behavior. The derived digital mobility parameters could serve as clinical outcomes for symptom monitoring, therapy management, rehabilitation, and fall risk assessment and prevention in patients with Parkinson’s disease.

## 6. Future Directions

In patients with PD and falls, an awareness of treatment-resistant symptoms such as fear of falling, postural instability, and FoG should be included in the clinical examination (e.g., by using the Hoehn and Yahr staging or the FoG questionnaire). Therapies should not only address factors that contribute to falls, but should also be appropriate to the stage of the disease. More RCTs are therefore needed that look at successful approaches at different stages of the disease, and that include the influence of co-morbidities such as cognitive decline and mood. The limited access to advanced dual-task therapies underlines the need not only to investigate them, but also to provide suitable therapy approaches that are available to a broader community. Several ongoing VR-TT studies looking at different paradigms and influencing factors are expected to provide new insights in the near future (for the study protocols, see [60,61]). Long-term studies of relevant outcome parameters that reflect not only patients’ ability in the clinical setting but also their performance in daily life are therefore needed. Wearables that analyze endurance, performance, and the complexity of locomotion reflect real-life performance and can be helpful in investigating relevant outcomes. The clinical implementation of wearable devices is ongoing and may even be included in closed-loop models, when deep brain stimulation is applied in the future [16]. Shoe-integrated sensors detecting FoG and providing consecutive sensory cueing are already available, which offers further perspectives for the treatment of falls in PD patients with FoG [81].

## Figures and Tables

**Figure 1 brainsci-14-00625-f001:**
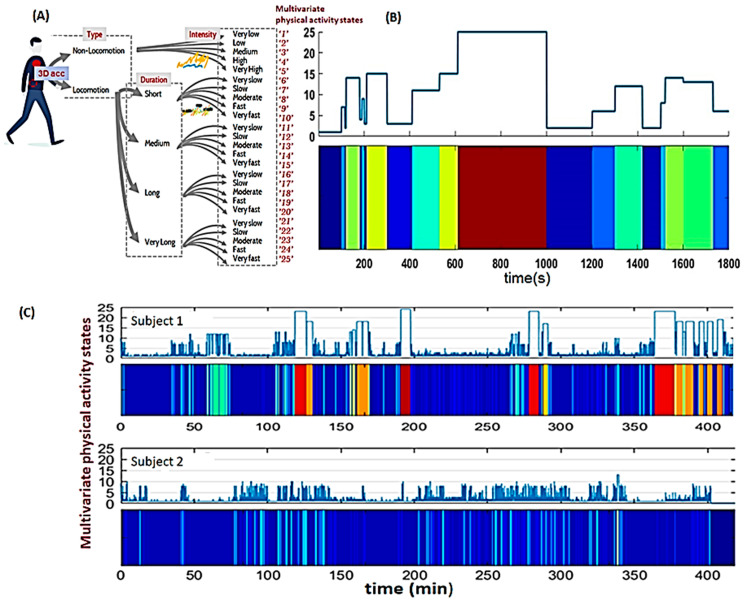
Illustrative representation of the concept of physical activity complexity: this approach involves combining multiple dimensions of physical activity, such as type, duration, and intensity, into 25 multivariate states (unitless). These 25 states quantify levels of physical activity from low to high. For example, state 1 corresponds to sedentary behavior, whereas state 25 represents long periods of walking at a high cadence. (**A**) is used to map the raw acceleration signal into a self-explanatory time-series that can be further analyzed and visually represented (**B**). Panel (**C**) contains illustrative examples of multivariate time-series recorded in two subjects with different levels of functioning. It can be observed that the features differentiating them reside in the diversity of multivariate states and in the frequency/dynamics of changes between states. The combination of these features corresponds to the mathematical definition of ‘complexity’. However, the abstraction of this concept can be intuitively interpreted clinically: higher/increased complexity is a marker of a high functioning level in terms of movement flexibility, the ability to achieve tasks, performance, and a variety of activities; conversely, decreased complexity may indicate the onset of functional decline in terms of a reduced range of mobility (rigidity), low endurance, and performance.

**Table 1 brainsci-14-00625-t001:** Relevant factors contributing to falls and diagnostic approaches (selection) in aging and in patients with PD.

	Symptom	Assessments (Selection)	References
**Aging related**	CDC approach		[3,4]
	Polypharmacy		[20]
	Osteoporosis		
	Vision		
	Mental health		
	Gait		
**Frailty**	Weakness	frailty phenotype	[5]
	Low physical activity		
	Low gait speed		
	Unintended weight loss		
	Self-reported exhaustion		
	Morbidity	frailty index	[21]
**Motor symptoms**	Bradykinesia	MDS-UPDRS III	[22]
	Motor fluctuations	MDS-UPDRS IV	[22]
		movement diary	
		WOQ-9	[23]
	Freezing of gait	FoG questionnaire	[24]
	Postural instability	Hoehn and Yahr stage	[22]
	Posture		
	Gait and balance	Mini-BESTest	[25]
**Non-motor symptoms**	Non-motor fluctuations	MDS-NMSS	[26]
	Orthostatic hypotension	tilt table testing	[27]
	Cognitive deterioration	MoCA	[28]
	Depression	HADS	[29]
	Anxiety	HADS	[29]
	Fear of falling	Falls Efficacy Scale-International (FES-I)	[30]
	Chronic pain	PD-PCS, KPPS	[31,32]
**Indirectly PD-related factors**	Polyneuropathy	neurography	[33]
		cobolamin testing	[34]
	Osteoarthrosis	radiography	
	Lumbar column degeneration	radiography, MRI	[11,35]

Abbreviations: CDC: Centers for Disease Control and Prevention; MDS-UPDRS: Movement Disorder Society-sponsored Unified Parkinson’s Disease Rating Scale; WOQ-9: Wearing off questionnaire-9; FoG: Freezing of Gait; NMSS: Non-motor symptom scale; MoCA: Montreal Cognitive Assessment; HADS: Hospital Anxiety and Depression Scale; FES-I: Falls Efficacy Scale-International; PD-PCS: Parkinson Disease Pain Classification System; KPPS: King’s Parkinson’s disease Pain Scale.

**Table 2 brainsci-14-00625-t002:** Selected studies investigating gait and balance in patients with PD.

References/Participants	Intervention	Control Group	Design	Primary Outcome Secondary Outcomes	Results
**Schenkman M et al. [49]** N1 = 43 N2 = 45 N3 = 40 Hoehn and Yahr stage ≤ 2 (within 5 years of diagnosis)	**high-intensity TT or** **moderate-intensity TT** four times/week for 6 months 80 or 60% of the maximal heart rate	wait list	single-blind RCT	**UPDRS III** V0_2_max	significant difference at 6 months between high intensity change 0.3 and wait list change 3.2 but not compared to moderate intensity change 2.0 V0_2_max improved in high intensity group compared to usual care
**van der Kolk NM et al. [50]** N1 = 65 N2 = 65 Hoehn and Yahr stage ≤ 2	**at-home home-trainer aerobic exercise** at 50–70% of heart rate reserve 3 times 30–45 min a week for 6 months (with exergaming experience, motivational app, and remote supervision)	at-home stretching (active control group, motivational app, and remote supervision)	single-blind RCT	**MDS-UPDRS III**	Med off score group difference 3.5 after 6 months
**Strouwen C et al. [51]** N1 = 56 N2 = 65 Hoehn and Yahr stage 2–3	**physiotherapeutic controlled gait training together or apart from cognitive tasks** for 6 weeks	control period without training (after 6 weeks before starting)	single-blind RCT	**dual-task gait velocity** 12-week follow-up	dual-task gait velocity improvement in both groups compared to the control period after intervention and at the 12-week follow-up
**Walton CC et al. [52]** N1 = 20 N2 = 18 PD patients with FoG	**cognitive training** (CT) twice a week for 7 weeks	active control group	double-blind RCT	**FoG during TUG** end of study (Med on and Med off condition)	percentage of FoG during TUG improved for CT at Med on improved processing speed improved daytime sleepiness
**Mirelman A et al. [53]** Elderly with falls N1 = 146 N2 = 136 Subgroup of PD patients: N1 = 66 N2 = 64 Hoehn and Yahr stage ≥ 3 ?	**VR-TT** 45 min three times/week for 6 weeks	TT alone	single-blind RCT	**falls 6 months before and after intervention** Cognition Gait Mobility Quality of life	all participants: significant reduction in the VR-TT group 11.92 vs. 6.0; not significant reduction in the TT group 10.71 vs. 8.27 PD patients: significant reduction in the VR-TT group 18.26 vs. 8.06; not significant reduction in the TT group 19.23 vs. 16.48 improvements in both groups improvements in both groups with more stable responses in the VR-TT after 6 months
**Kwok JYY et al. [54]** N1 = 71 N2 = 67 Hoehn and Yahr stage 2–3	**Mindfulness Yoga** 90 min group for 8 weeks	stretching and resistant training exercise 60 min group	single-blind RCT	**HADS** 8 weeks (T1) 12 weeks (T2) MDS-UPDRS III TUG HRQOL	significant time x group interaction anxiety: T1= −1.79 T2 = −2.05 depression: T1 = −2.75 T2 = −2.75 T1 = −5.19 T2 = −4.71 ns T1 = −7.7 T2 = −7.99
**Li G et al. [55]** N1 = 143 N2 = 187 Hoehn and Yahr stage ≤ 2.5	**Thai Chi** 60 min twice/week for 4.3 years	control group without any intervention	unblinded	**UPDRS** 4.3 years UPDRS III LEDD	between group differences −2 −2 −233 mg
**Capato TTC et al. [56]** N1 = 17 N2 = 18 Hoehn and Yahr stage 4	**multimodal balance training and rhythmic auditory stimuli (RAS)** twice/week for 5 weeks	multimodal balance training	single-blind RCT	**Mini-BESTest** 1 month 6 months	improvement in both groups after training and 1 month after training improvement after 6 months in the RAS group only

Abbreviations: TT: treadmill training; MDS-UPDRS: Movement Disorder Society-sponsored Unified Parkinson’s Disease Rating Scale; VR: Virtual Reality; FoG: Freezing of Gait; TUG: Timed up and go; HADS: Hospital Anxiety and Depression Scale; HRQOL: Health-related Quality of Life; LEDD: L-dopa equivalent daily dosage.

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
