# Peer review of "Imbalance and Falls in Patients with Parkinson’s Disease: Causes and Recent Developments in Training and Sensor-Based Assessment"

_brainsci, 2024, doi:10.3390/brainsci14070625_

Round 1

Reviewer 1 Report

Comments and Suggestions for Authors

I found this manuscript interesting for readers and comprehensive. I have onle few minor comments:

Page 2, Section 2, line 5. Sentence “In PD patients with falls, >2 falls in the past year, the presence of motor fluctuations, an activities of daily living (ADL) Unified PD Rating Scale (UPDRS) score >12, Levodopa equivalent daily doses (LEDD) >700 mg, and a Berg balance scale score of<49 points predict subsequent falls with moderate to high accuracy [4].”

Which section of UPDRS scale – presumably UPDRS-III (or total score) is presented?

Section 2.1. Sentence ”Postural deformities further contribute to imbalance by changing the center of gravidity.” Shouldn’t it be "center of gravity"?

Figure 1. What is the measuring unit of the “multivariative physical activity states” in the graphs B and C?

Author Response

Reviewer 1

I found this manuscript interesting for readers and comprehensive. I have only few minor comments:

We thank the reviewer of her/his comment.

Page 2, Section 2, line 5. Sentence “In PD patients with falls, >2 falls in the past year, the presence of motor fluctuations, an activities of daily living (ADL) Unified PD Rating Scale (UPDRS) score >12, Levodopa equivalent daily doses (LEDD) >700 mg, and a Berg balance scale score of<49 points predict subsequent falls with moderate to high accuracy [4].” 

Which section of UPDRS scale – presumably UPDRS-III (or total score) is presented?

We changed the order of the phrase to make clear that the study refers to the ADL part of the UPDRS (part II) to: Unified PD Rating Scale (UPDRS) activities of daily living (ADL) score >12

Section 2.1. Sentence ”Postural deformities further contribute to imbalance by changing the center of gravidity.” Shouldn’t it be "center of gravity"?

We thank the reviewer for this important correction.

Figure 1. What is the measuring unit of the “multivariative physical activity states” in the graphs B and C?

The multivariate physical activity states are unitless. However, it is worth mentioning that the numbers assigned to the respective states (ranging from 1 to 25) quantify levels of physical activity from low to high. For example, state 1 corresponds to sedentary behavior, whereas state 25 represents long periods of walking at a high cadence. We have included this information in the revised version of the manuscript (caption Figure 1).

Reviewer 2 Report

Comments and Suggestions for Authors

The article titled "Imbalance and Falls in Patients with Parkinson Disease: Causes and Recent Developments in Training and Sensor-Based Assessment" provides a comprehensive review of the factors contributing to balance issues and falls in patients with Parkinson's disease (PD), along with the latest advancements in training methodologies and sensor-based assessments. The authors discuss the significant impact of both motor and non-motor symptoms of PD on patients' mobility and highlight the effectiveness of various training techniques, including treadmill training in non-immersive virtual reality, which has shown promise in reducing falls.

The topic is crucial as falls significantly deteriorate the quality of life and increase the mortality risk among PD patients. Understanding the multifactorial causes of falls and improving assessment and intervention strategies can significantly enhance patient care. This article is especially important for clinicians and researchers in developing targeted therapies that mitigate fall risks and improve overall patient outcomes.

Note on Methodology:

While the article provides an extensive overview of the current knowledge and recent developments in managing imbalance and falls in PD, it appears to compile evidence from a broad spectrum of sources rather than following a systematic review methodology. This approach may limit the ability to perform a meta-analysis but allows for a comprehensive discussion of a wide range of interventions and assessments, capturing diverse aspects of the problem.

Author Response

We thank the reviewer of her/his comments.

Reviewer 3 Report

Comments and Suggestions for Authors

The purpose of this review was to summarizes potential causes for imbalance in patients with PD along with recent advances in mobility assessment using wearable devices, gait training and virtual reality-based gait training.  

 Overall, I found this review article to be well-written, easy to follow, and a good synopsis of the literature on the topic. It adds to the literature on the topic as I don’t think that there is a review on quite the same topic. I think it will be of interest to readers of the journal and researchers in many adjacent fields.

 I don’t think the viewpoint article has any major flaws or weaknesses. Therefore, I have only minor comments in regard to formatting and typographical error issues.

 Minor comments:

1. For some reason, there are no line numbers in the document, which makes it hard to point to lines of text easily when writing the review to point out issues. Not sure if the authors used the correct format to submit the paper, such as the word template provided by the journal.

2. Relatedly, in the 3rd line of the Introduction there are semicolons between the refrence numbers and the references do not start at 1. I don’t think this is in line with the journal format.

3. Table 2 has rather big spaces between some words and text. Maybe it could be reformatted to improve how it looks.

4. Although over all very well written there are some typos. As a few examples, in the paragraph right before Section 2.2 by citation 6 the sentence has 2 periods. Second paragraph of Section 2.2 has no period. In the paragraph after Table 2 there is no space after UPDRS and before III. There are more examples but please proofread.

5. Some sections have really choppy paragraphs where they could have been combined into larger paragraphs. Such as Section 2.2 had several paragraphs of only 1-2 sentences each.

6. It appears that there are formatting inconsistencies in the Bibliography. Many times the article title does not have all the first words capped in some references but does in others. For instance, just one example is ref 6 and 8 have caps for first words and 11 and 12 for instance do not. Many examples of this for numerous references.

Comments on the Quality of English Language

minor proofreading needed some typos

Author Response

Reviewer 3

The purpose of this review was to summarizes potential causes for imbalance in patients with PD along with recent advances in mobility assessment using wearable devices, gait training and virtual reality-based gait training.  

 Overall, I found this review article to be well-written, easy to follow, and a good synopsis of the literature on the topic. It adds to the literature on the topic as I don’t think that there is a review on quite the same topic. I think it will be of interest to readers of the journal and researchers in many adjacent fields.

 I don’t think the viewpoint article has any major flaws or weaknesses. Therefore, I have only minor comments in regard to formatting and typographical error issues.

We thank the reviewer of her/his comments.

 Minor comments:

  1. For some reason, there are no line numbers in the document, which makes it hard to point to lines of text easily when writing the review to point out issues. Not sure if the authors used the correct format to submit the paper, such as the word template provided by the journal.

The current form was provided by the journal.

  1. 2.Relatedly, in the 3rdline of the Introduction there are semicolons between the reference numbers and the references do not start at 1. I don’t think this is in line with the journal format.

The style may be changed by the type setting or the proof-reading service of the journal if necessary.

  1. Table 2 has rather big spaces between some words and text. Maybe it could be reformatted to improve how it looks.

We improved table 1 and 2 accordingly.

  1. Although over all very well written there are some typos. As a few examples, in the paragraph right before Section 2.2 by citation 6 the sentence has 2 periods. Second paragraph of Section 2.2 has no period. In the paragraph after Table 2 there is no space after UPDRS and before III. There are more examples but please proofread.

We corrected the typos.

  1. Some sections have really choppy paragraphs where they could have been combined into larger paragraphs. Such as Section 2.2 had several paragraphs of only 1-2 sentences each.

We now summarize the items in one paragraph a suggested.

  1. It appears that there are formatting inconsistencies in the Bibliography. Many times the article title does not have all the first words capped in some references but does in others. For instance, just one example is ref 6 and 8 have caps for first words and 11 and 12 for instance do not. Many examples of this for numerous references.

We hope that the typesetting will further